# Diagnostic Differentiation between Pancreatitis and Pancreatic Cancer: A Scoping Review

**DOI:** 10.3390/diagnostics14030290

**Published:** 2024-01-29

**Authors:** Fusi Madela, Lucien Ferndale, Colleen Aldous

**Affiliations:** Department of Surgery, School of Clinical Medicine, College of Health Sciences, University of KwaZulu-Natal, Durban 4000, South Africa; lucienferndale@gmail.com (L.F.);

**Keywords:** pancreatitis, chronic pancreatitis, pancreatic adenocarcinoma, pancreatic carcinoma, pancreatic cancer

## Abstract

Pancreatitis, encompassing acute and chronic forms, and pancreatic cancer pose significant challenges to the exocrine tissue of the pancreas. Recurrence rates and complications following acute pancreatitis episodes can lead to long-term risks, including diabetes mellitus. Chronic pancreatitis can develop in approximately 15% of cases, regardless of the initial episode’s severity. Alcohol-induced pancreatitis, idiopathic causes, cigarette smoking, and hereditary pancreatitis contribute to the progression to chronic pancreatitis. Chronic pancreatitis is associated with an increased risk of pancreatic cancer, with older age at onset and smoking identified as risk factors. This scoping review aims to synthesise recent publications (2017–2022) on the diagnostic differentiation between pancreatitis and pancreatic cancer while identifying knowledge gaps in the field. The review focuses on biomarkers and imaging techniques in individuals with pancreatitis and pancreatic cancer. Promising biomarkers such as faecal elastase-1 and specific chemokines offer non-invasive ways to assess pancreatic insufficiency and detect early biomarkers for chronic pancreatitis. Imaging techniques, including computed tomography (CT), magnetic resonance imaging (MRI), endoscopic ultrasound (EUS), and positron emission tomography (PET), aid in differentiating between chronic pancreatitis and pancreatic cancer. However, accurately distinguishing between the two conditions remains a challenge, particularly when a mass is present in the head of the pancreas. Several knowledge gaps persist despite advancements in understanding the association between pancreatitis and pancreatic cancer, including the correlation between histopathological grading systems, non-invasive imaging techniques, and biomarkers in chronic pancreatitis to determine the risk of progression to pancreatic cancer, as well as differentiating between the two conditions. Further research is necessary to enhance our understanding of these aspects, which can ultimately improve the diagnosis and management of pancreatitis and pancreatic cancer.

## 1. Introduction

Acute and chronic pancreatitis (CP), together with pancreatic cancer (PC), account for a significant proportion of diseases affecting the exocrine tissue of the pancreatic gland [1]. A nationwide study of acute pancreatitis (AP) in Japan showed a recurrence rate of 20% and a complication rate with diabetes mellitus of up to 54% following index AP [2]. The same study demonstrated a transition to CP in up to 15% of cases, with no difference in transition rates based on the severity stratification of AP [2]. A different study showed that recurrent acute pancreatitis (RAP) progresses to CP, with progressive acute pancreatitis (PAP) accounting for 24% of AP [3]. A retrospective cohort study in Beijing demonstrated that independent risk factors for progression to CP included more than four episodes of RAP, idiopathic pancreatitis, and pseudocysts [4].

This progression is worse with alcohol-induced pancreatitis (48%), idiopathic causes (47%), cigarette smoking, and hereditary pancreatitis than other aetiologies [3]. In the Japanese study, the transition rate was found to be significantly higher in the alcohol-induced AP cohort compared to all other causes [2]. Alcohol and cigarette use increase the susceptibility of AP to recurring episodes by reducing the threshold for triggering trypsinogen activation in acinar cells, hindering the secretion of pancreatic duct cells, or affecting the immune system, resulting in chronic inflammation. This is thought to be due to the failure of protective mechanisms in the normal healing process, such as DNA repair, apoptosis, and the immune-mediated elimination of dysplastic cells [5]. The overall mortality in the AP cohort is worse in the PAP than in the nonprogressive acute pancreatitis cohorts [3].

Several risk factors for PC have been identified, including cigarette smoking, alcohol use, diabetes mellitus, and CP (see Figure 1 for the risk factors and relative risk) [6,7]. The latter is responsible for up to a 16-fold increased risk of PC [8]. In one study, Korpela et al. found the incidence of PC in CP cohorts to be 6.6%, with older age at onset and smoking as the risk factors. They also found features of CP in 38.8% of histopathological specimens following surgery for PC [9]. The classic histopathological features used to diagnose CP include fibrosis, acinar atrophy, and ductal abnormalities. However, there are no specific distinguishing features for different aetiologies of CP based on histopathology, which underscores the significance of correlation with clinical and radiological findings. Furthermore, there is no consensus on the histopathological grading system for the severity of CP. Some pathologists classify CP into mild, moderate, and severe CP using the fibrosis scoring system proposed by Klöppel and Mallet in 1991 [10].

The average lag period between CP and PC is reported to be about 20 years [1]. However, there is a linear relationship between the proportion of CP patients developing PC over time, with 1.8% of patients diagnosed with PC after 10 years of diagnosis with pancreatitis and 4% after 20 years [11].

It is still unclear how much of this observed risk factor of CP is confounded by cigarette smoking and alcohol use, both of which are independent risk factors for PC [8]. Even though 70% of cases of CP are attributed to alcohol abuse, a large majority of alcohol users (95%) never suffer from CP [5].

The relationship between CP and PC is further illustrated by the observation that a significant proportion of PC (~5%) is misdiagnosed as CP, leading to potentially poorer outcomes occasioned by delay in diagnosis [8]. There are usually diagnostic difficulties between PC and CP in the case of a mass in the head of the pancreas due to the similar features of a hard mass, vascular invasion, or adjacent organ invasion seen in both diseases. The incidence of malignancy in CP patients with an apparent inflammatory mass in the head of the pancreas may be as high as 33.7% [12]. Epidemiology studies in the United States have demonstrated an overlap between patients presenting with recurrent acute pancreatitis and the development of CP and PC [13].

This association between CP and PC follows other patterns of association between chronic inflammation and subsequent malignancies [14]. Ninety percent of PCs have *K-ras* oncogene mutations, which are also found in CP sufferers, suggesting a likely common pathophysiology between the two diseases [8].

This review aims to qualitatively synthesise recent literature on the preoperative diagnostic differentiation between pancreatitis and PC and identify potential knowledge gaps.

## 2. Methodology

### 2.1. Search Strategy

We searched four online databases (Cochrane, PubMed, Web of Science, and Google Scholar) plus grey literature. The combination of medical subject headings (MeSH) used in the search was “pancreatitis”, or “chronic pancreatitis” and “pancreatic cancer” or “pancreatic carcinoma” or “pancreatic adenocarcinoma” appearing in the title of the publication. We used the Boolean operators (‘AND’ and ‘OR’) to combine keywords effectively and applied filters to limit the search results to the years 2017 to 2022. The search criteria were modified post hoc to focus on preoperative diagnostic differentiation between pancreatitis and pancreatic cancer and were applied to all citations for assessment of relevance. The citation lists obtained were added to a reference manager, and an analysis of text words appearing in the title and abstracts was initially conducted.

PDF copies of all the references were uploaded, and duplicates were removed. Nvivo^R^ 12 version 1.7.1 (Lumivero) was used to code the literature under the following headings and subheadings.

### 2.2. Study Type


Case reports.Randomised controlled studies.Cohort studies.Systematic reviews and meta-analyses.


### 2.3. Diagnostic Differentiation


Biomarkers.Imaging.


### 2.4. Eligibility Criteria

#### 2.4.1. Inclusion Criteria

All publications from 2017 to October 2022, including case reports, cohort studies, randomised controlled studies, and systematic reviews and meta-analyses, with a combination of MeSH appearing in the title of the publication, were included.

#### 2.4.2. Exclusion Criteria

Animal studies, non-English language publications, e-posters, conference reports, letters, supplements, textbook chapters, or theses were excluded.

The literature search findings are reported using a flow diagram following the guidelines of the Preferred Reporting Items for Systematic Reviews and Meta-analyses extension for scoping reviews (PRISMA-ScR) [15]. The protocol of the scoping review has been registered on Open Science Framework registries https://doi.org/10.17605/OSF.IO/UVHZX accessed on 12 February 2023.

## 3. Results

Figure 2 represents a comprehensive flow chart outlining the outcomes of our literature search. A total of 299 articles were initially identified from the Pubmed, Cochrane, Google Scholar, and Web of Science databases. After merging records and removing duplicates, the abstracts of the remaining articles were screened to determine their relevance. Among these, 176 full-text articles were assessed for eligibility, and 58 were excluded based on the predefined exclusion criteria. Ultimately, 118 articles were included in the study for qualitative synthesis. Among the included articles, there were 75 cohort studies, 31 reviews/meta-analyses, and 12 case reports. Notably, no randomised controlled trials were identified during the search process.

### 3.1. Data Analysis and Presentation

#### 3.1.1. Diagnostic Differentiation

There were 92 out of 118 articles that covered the subject of diagnosis with overlap across the two subheadings coded for diagnosis, namely, “Biomarkers” (in 60 articles out of 92) and “Imaging” (46 out of 92).

#### 3.1.2. Biomarkers

Data charting tool included:Authors.Year of publication.Country of origin.Biomarkers identified for diagnostic differentiation.

The absolute number of publications that presented biomarkers were as follows:

Ca 19.9 (23), IgG4^+^/IgG (18), Cytokines and chemokines (13), CEA (6), and Enzymes, including faecal elastase-1 and amylase (3). Figure 3 summarises biomarkers published for diagnostic differentiation in the period under review.

See Table 1 for the biomarker data extracted from each publication.

#### 3.1.3. Imaging

Data charting tool included:Authors.Year of publication.Country of origin.Cross-sectional imaging identified for diagnostic differentiation.

The absolute number of publications that presented on cross-sectional imaging were as follows: CT/enhanced CT radiomics (31), EUS/CEUS/EUS FNA/EUS FNB (20), MRI/MRCP/MR elastography or tomoelastography (16), FDG PET (8), ERCP (7), TAUS (3), and Infrared spectroscopy (1). Figure 4 summarises imaging types published for diagnostic differentiation in recent publications.

See Table 2 for the imaging data extracted from each publication.

## 4. Discussion

We searched recent publications to establish if there are new studies on biomarkers and imaging that may help with the preoperative diagnostic differentiation between CP and PC. Although there are many studies on Ca 19-9 as a biomarker of CP and PC, there is still uncertainty with respect to sensitivity and specificity regarding its use. Ca 19-9 is commonly used for the diagnosis of PDAC, but its routine use is not recommended in patients with CP due to its low specificity. Faecal elastase-1 and specific chemokines offer non-invasive ways to assess pancreatic insufficiency and detect early biomarkers for CP. There are a few novel biomarkers that hold promise but more research is still needed.

Direct markers of pancreatic exocrine function are invasive because they involve obtaining pancreatic juices via endoscopy or Dreiling tube following stimulation by secretin or cholecystokinin. Bicarbonate, lipase, or trypsin is then measured, and these measurements are highly sensitive for late CP but have a lower sensitivity range of 70–75% for early chronic pancreatitis [25]. Indirect markers, on the other hand, are non-invasive tests of pancreatic insufficiency. One example is faecal elastase-1, with a cutoff of 100 micrograms having a sensitivity of 46.5% and a specificity of 88% for the diagnosis of CP [25]. Early biomarkers for CP include chemokines like transforming growth factor-Beta 1 (TGF- β1), platelet-derived growth factor BB, and chemerin, which are elevated in CP [25]. Other reported markers are des-Leu albumin, which is present in 68% of CP, and YKL-40, a mammalian chitinase-like protein, which is elevated in CP [25].

The serologic marker for autoimmune pancreatitis (AIP) is immunoglobulin G4 (IgG4). Its elevation supports the diagnosis of AIP, but normal levels do not exclude it. It can also be elevated in 10% of patients with pancreatic cancer [24,57].

There are no highly sensitive nor specific tumour markers for PC [25]. Although serum carbohydrate antigen 19-9 (Ca 19-9) is commonly used for the diagnosis of pancreatic ductal adenocarcinoma (PDAC), its routine use is not recommended in the cohort of CP patients due to its low specificity because inflammation is found in both conditions [72]. The reported sensitivity and specificity of Ca 19-9 are 78% and 83%, respectively, while those of carcinoembryonic antigen (CEA) are 44% and 85%, respectively [25,36].

Pancreatic stellate cells are the main cells involved in the fibrosis observed in PDAC and CP by activating the *α*-smooth muscle actin (*α*SMA), which is strongly expressed in PDAC and moderately in CP when contrasted to the weak expression observed in healthy individuals. This immunoexpression of the *α*SMA protein is higher in larger tumours and higher grades of differentiation of tumours [40].

There are other novel biomarkers for PC with promising early results like tissue inhibitor of metalloproteinase 1 (TIMP-1), matrix metalloproteinase 9 (MMP-9), urokinase-type plasminogen activator receptor (uPAR), osteopontin, heat shock protein 70 (HSP 70), and macrophage inhibitor cytokine (MIC-1) [25,36].

Imaging modalities such as CT, MRI, EUS, and PET/CT are used to distinguish between CP and PC. EUS-guided FNB was found to have higher diagnostic accuracy than EUS-guided FNA for differentiating pseudo-tumour-like pancreatitis from PC. CEH-EUS improves specificity when differentiating focal AIP from PC, and FDG PET has a higher sensitivity for AIP than for PC. It is important to differentiate between PC and CP for early detection and appropriate management.

Korpela et al. recommend that patients with a biliary stricture and other risk factors for PC, including higher age at onset of CP, should undergo assessment with computed tomography (CT) and endoscopic ultrasound (EUS) [9]. Several imaging features may help distinguish CP from PC on CT and magnetic resonance imaging (MRI). A hypovascular mass associated with a smooth dilatation of an upstream pancreatic duct and parenchymal atrophy is more suggestive of PC as opposed to an irregular pancreatic duct, and it is a penetrating duct sign that is more in keeping with focal chronic pancreatitis [80]. A double duct sign is nonspecific for PC, as it does occur in benign pathologies like choledocholithiasis and CP. However, the common bile duct stricture in the head of the pancreas tends to be longer and tapered in CP versus the short, abrupt cutoff stenosis observed in PC [80]. Isolated pancreatic duct dilatation has a 35% higher probability of being PC in the absence of CP [80]. The presence of a mass on CT or MRI in chronic calcifying pancreatitis associated with a dilated common bile duct (CBD) is suggestive of a malignancy [89].

Diffusion-weighted imaging and apparent diffusion coefficient values alone cannot distinguish PC from CP [80]; however, combined detection sensitivity and specificity of CT and Diffusion-Weighted Imaging–Magnetic Resonance Imaging (DWI-MRI) with Magnetic Resonance Cholangiopancreatography (MRCP) for mass-forming pancreatitis and PC is higher than either modality on their own [76].

Compared to conventional ultrasound, contrast-enhanced ultrasound time–intensity curves (TICs) with SonoVue^®^ [bracco imaging] contrast is better at distinguishing pseudo-tumour-like pancreatitis from PC [94].

Contrast-enhanced high mechanical index EUS has a higher sensitivity and specificity (96% and 91%, respectively) to discriminate CP from PDAC than B-mode EUS (92% and 63%, respectively), endoscopic sonoelastography (96% and 38%, respectively), and multidetector contrast-enhanced CT (89% and 70%, respectively) [82]. EUS-guided fine-needle biopsy (FNB) has higher diagnostic accuracy and sensitivity for differentiating pseudo-tumour-like pancreatitis from PC than EUS-guided fine-needle aspiration (FNA) [61]. A recent retrospective study suggested that EUS should still form part of the diagnostic algorithm for evaluating acute, idiopathic, or CPof an unclear cause even when CT/MRI does not show a mass lesion because there is still a 5.3% risk of PC [93].

The combination of endoscopic retrograde cholangiopancreatography (ERCP) with tumour markers (Ca 19-9 and CEA) improves the diagnostic accuracy and sensitivity of PC from pseudo-tumour-like pancreatitis, thereby lowering the rate of a missed diagnosis of PC [44].

Even though imaging signs overlap between AIP and PC, CT values for AIP are significantly higher than those for PC [77]. Where there is a pancreatic mass, typical features of PC should be excluded on CT or MRI, those typical features being hypoattenuating lesion, pancreatic parenchymal compression by mass, abrupt cutoff of the dilated pancreatic duct with distal atrophy of the gland, vessel involvement, double-duct signs, and lymphadenopathy [41]. The CT values in an AIP cohort were statistically higher than in a PC cohort [77]. PDAC has a significantly higher stiffness and fluidity than AIP and healthy patients on MRI tomoelastography [83]. Although AIP resembles PC on cross-sectional imaging, EUS may differentiate between the two pathologies owing to peripancreatic hypoechoic margins (PHMs), which are present in 40% of focal AIP1 patients but not seen in PC [81]. The pancreatic duct wall was thickened in 67% of a focal AIP1 cohort compared to 6.7% of a PC cohort [81]. Contrast-enhanced harmonic endoscopic ultrasound (CEH-EUS) improved specificity when differentiating focal AIP from PC [79].

Studies on AIP and PC show that cross-sectional imaging using early [18] F-FDG PET/CT scans (PET_60min_) has a predominant focal metabolic avidity with a higher average uptake (SUVmax of 7.30 ± 3.21) in PC when compared to a predominantly diffuse pattern of avidity in AIP with an average SUVmax of 5.24 ± 1.81. Just over 50% of cases in PC tend to have pancreatic duct dilatation, which is only observed in a quarter of cases in AIP [75]. FDG PET has a higher sensitivity for AIP than for PC [68].

## 5. Conclusions

A review of research on pancreatitis and PC in the last five years demonstrates extensive exploratory work. However, much is unknown and poorly understood regarding their association with regard to disease progression from pancreatitis to PC, and there is still a significant challenge in differentiating between the two entities. A misdiagnosis of pancreatitis as PC is accompanied by high morbidity resulting from inappropriate management, and a missed diagnosis of PC is accompanied by mortality due to the delay in management. Some of the areas needing further exploration are the correlation between the histopathological grading systems with non-invasive imaging and biomarkers of CP and whether the risk of progression to PC in CP cohorts is associated with a specific grading or severity scoring. Because of the low sensitivity and specificity in the current imaging modalities and biomarkers in use, there is a need for further research on radiomics, metabolomics, PC cytokines, and liquid biopsy to improve accuracy for both imaging modalities and biomarkers for diagnostic differentiation between pancreatitis and pancreatic cancer (see Figure 5 for a proposed diagnostic algorithm for pancreatitis with a head-of-pancreas mass).

## Figures and Tables

**Figure 1 diagnostics-14-00290-f001:**
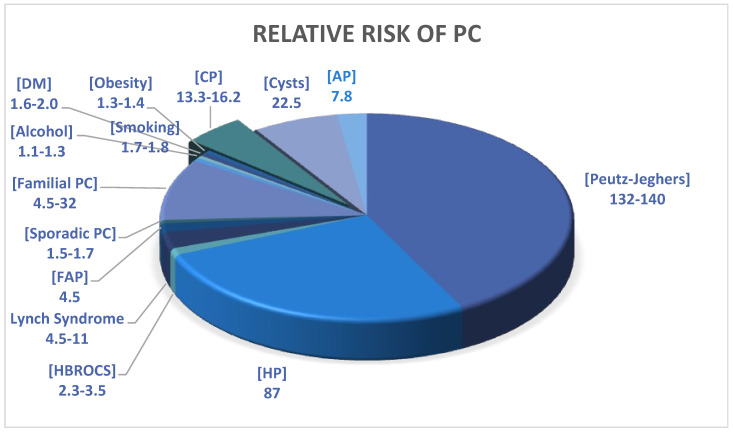
Relative risk for development of PC [6,7].

**Figure 2 diagnostics-14-00290-f002:**
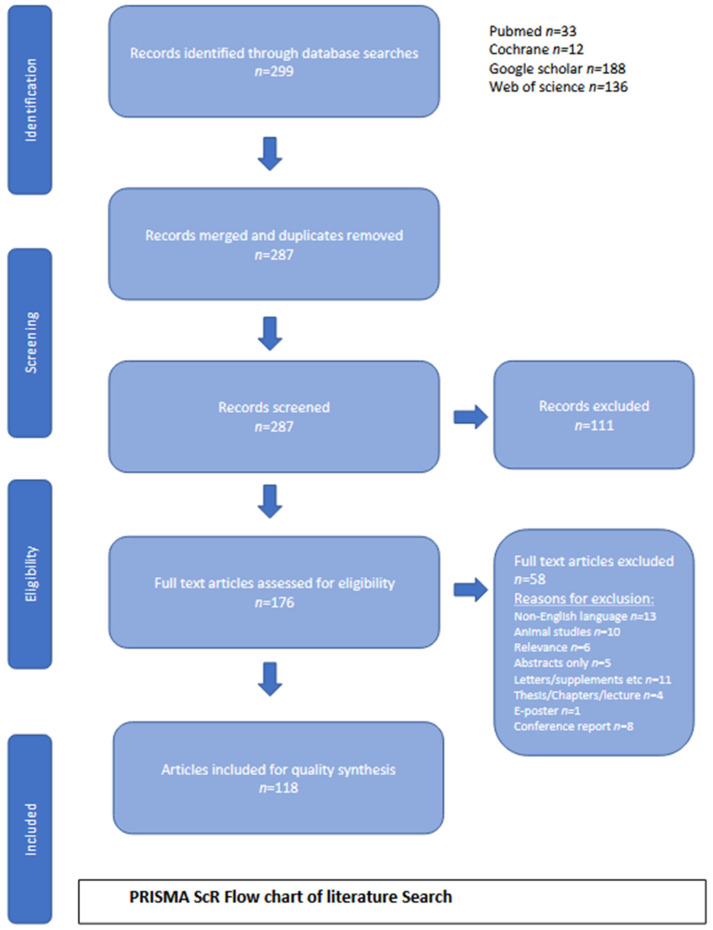
Prisma-ScR flow chart.

**Figure 3 diagnostics-14-00290-f003:**
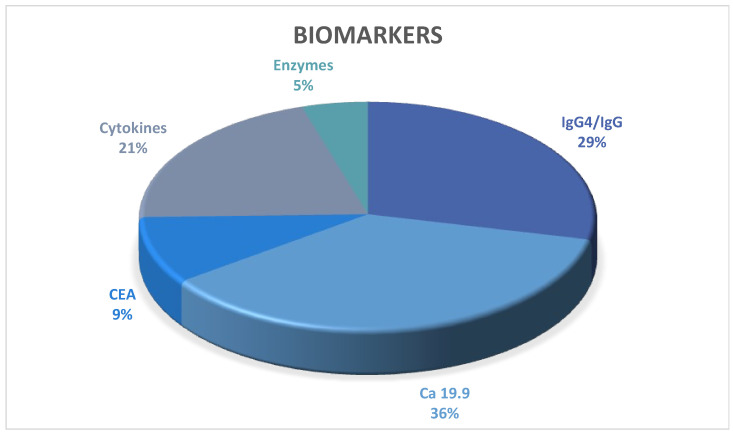
Biomarkers.

**Figure 4 diagnostics-14-00290-f004:**
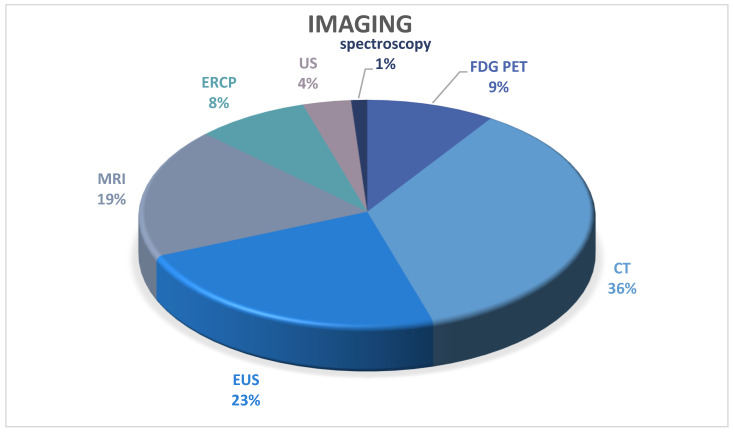
Imaging.

**Figure 5 diagnostics-14-00290-f005:**
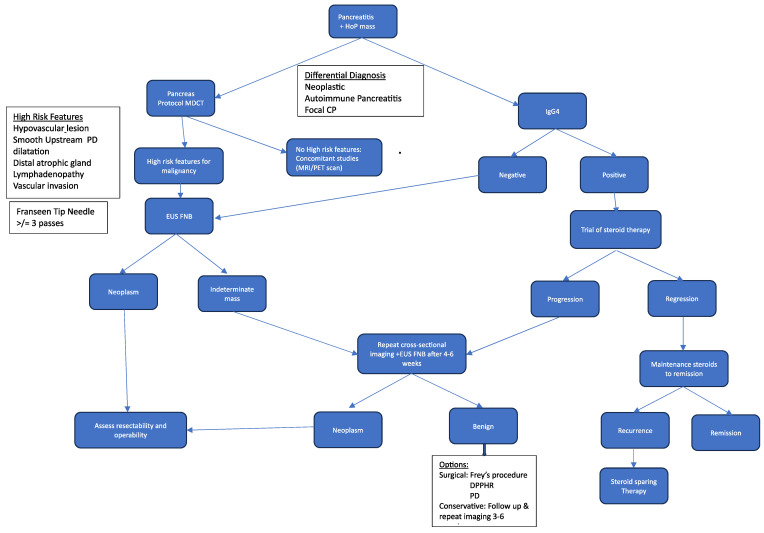
Proposed diagnostic algorithm for pancreatitis with head of pancreas mass.

**Table 1 diagnostics-14-00290-t001:** Studies included for review of biomarkers.

First Author	Year	Country	Biomarkers
1. Li G et al. [16]	2019	China	IgG4+/IgG+ plasma cell ratio > 40%
2. Tang D et al. [17]	2018	China	overexpression of galectin-1 promotes PSC activity
3. Haeberle L et al. [18]	2018	Germany	PDAC stroma > mucin content than CP
4. Dite P et al. [19]	2019	Czech Republic	Plasmatic IgG4 levels >135 mg/dL in PC
5. Jin G et al. [20]	2020	China	Pancreatic stellate cell (PSC)-stimulating factors
6. Marinho R et al. [21]	2019	Portugal	CA19-9 and IgG4
7. Negoi I et al. [22]	2019	Romania	CA 199 and glycosylation alterations
8. Aronen A et al. [23]	2017	Finland	P-suPAR was significantly higher in PC
9. Dai C et al. [24]	2018	China	IgG4 high specificity and low sensitivity (AIPvsPC)
10. Chou C et al. [25]	2020	Taiwan	use of mass spectrometry for protein biomarkers
11. Li W et al. [26]	2022	China	CA19-9 and *KRAS* mutations in blood
12. Huang C et al. [27]	2021	France	nuclear protein 1 (NUPR1
13. Park W et al. [28]	2020	USA	unique immune signature panels
14. Prokopchuk O et al. [29]	2018	Germany	matrix metalloproteinase inhibitor TIMP1
15. Li Z et al. [30]	2019	Germany	Interleukin-18
16. Macinga P et al. [31]	2017	Czech Republic	IgG4 levels
17. Hansen S et al. [32]	2021	Denmark	Low and high amylase is associated with PC and CP
18. Sanh N et al. [33]	2018	USA	transferrin, ER-60 protein, proapolipoprotein, tropomyosin 1, alpha 1 actin precursor, ACTB protein, and gamma 2 propeptide, aldehyde dehydrogenase 1A1, pancreatic lipase, and annexin A1
19. Chu C et al. [34]	2019	China	ratio of IgG4/IgG and CA 19-9
20. Yan T et al. [35]	2017	China	Ca19-9
21. Kandikattu H et al. [36]	2020	USA	Cytokines IL-4, IL-5, IL-6, IL-13, IL-15, IL-17, IL-18, IFN-γ, TNF-α, and chemokines
22. Ishikawa T et al. [37]	2020	Japan	serum IgG4 and CA19-9
23. Detlefsen S et al. [38]	2018	Denmark	anti-plasminogen binding peptide, anticarbonic anhydrase II, IgG4
24. Dranka-Bojarowska D et al. [39]	2020	Poland	MMP-2, MMP-9, CA19-9, and CEA
25. Winter K et al. [40]	2021	Poland	αSMA expression higher in tumours > than 3 cm
26. Lin T et al. [41]	2020	Taiwan	CA19-9, CEA, CRP and IgG4
27. Saraswat M et al. [42]	2017	Finland	Proteomics
28. Chou, O et al. [43]	2022	China	sodium–glucose cotransporter 2 inhibitors vs. dipeptidyl peptidase-4 inhibitors
29. Luo B et al. [44]	2019	China	serum CEA and CA19-9
30. Wen Y et al. [45]	2021	China	Macrophage Migration Inhibitory Factor
31. Sheng L et al. [46]	2021	China	serum exosomal microRNAs
32. Poddighe D [47]	2021	Kazakhstan	IgG4, IL-4, IL-5, IL-13, IL-10, and TGF-β
33. Dickerson L et al. [48]	2019	UK	IgG4 immunohistochemistry
34. Ghassem-Zadeh S et al. [49]	2020	Germany	Novel Autoantibody Signatures
35. Mungamuri S et al. [50]	2019	India	cytokines
36. Sunami Y et al. [51]	2022	Germany	scRNAseq and bioinformatics analyses
37. Lindahl A et al. [52]	2017	Sweden	Glycocholic acid, N-palmitoyl glutamic acid and hexanoylcarnitine
38. Gluszek S et al. [53]	2020	Poland	pentraxin 3 (PTX3)
39. Bang U et al. [54]	2018	Denmark	cytokines and chemokines
40. Zhao X et al. [55]	2020	China	CA19-9
41. Nissinen S et al. [56]	2021	Finland	polyamines—acetylputrescine, diacetylspermidine, N8-acetylspermidine and diacetylputrescine
42. Agarwal K et al. [57]	2022	USA	CA19-9, IgG4
43. Zhang H et al. [58]	2018	China	CA19-9
44. Kunovsky L et al. [59]	2021	Czech Republic	cytotoxin-associated gene A
45. Macinga P et al. [60]	2021	Czech Republic	IgG4
46. Korpela T et al. [9]	2020	Finland	CA19-9, CEA
47. Grassia R et al. [61]	2020	Italy	CA19-9
48. Zhou Q et al. [62]	2020	China	T Lymphocytes
49. Walling A et al. [63]	2017	USA	CA19-9
50. Rana S et al. [64]	2018	India	CA19-9
51. Hsu W et al. [65]	2018	Taiwan	IgG and IgG4 levels
52. Kalayarasan R et al. [66]	2021	India	Cytokines and chemokines
53. Jiang H et al. [67]	2018	Canada	CA19-9 and IgG4/IgG
54. Ohtani M et al. [68]	2021	Japan	CA19-9 and IgG4/IgG
55. Kim H et al. [69]	2022	South Korea	CA19-9
56. Li S et al. [70]	2017	China	CA19-9
57. Bieliuniene E et al. [71]	2019	Lithuania	Faecal elastase-1
58. Umans D et al. [72]	2021	Netherlands	CA19-9
59. Matsubayashi H et al. [73]	2021	Japan	CEA, CA19-9, IgG4
60. Miyoshi H et al. [74]	2019	Japan	CEA, CA19-9, IgG4

**Table 2 diagnostics-14-00290-t002:** Studies included for review of imaging.

First Author	Year of Publication	Country	Modalities
1. Zhang J et al. [75]	2017	China	18F-FDG PET/CT
2. Umans D et al. [72]	2021	Netherlands	CT, EUS, MRI
3. Jiang S et al. [76]	2021	China	CT and MRI
4. Korpela T et al. [9]	2020	Finland	CT, MRCP, US, FDG PET/CT, EUS
5. Zhao Y et al. [77]	2021	China	CT
6. Ergin E et al. [78]	2021	Turkey	CT, ERCP
7. Agarwal K et al. [57]	2022	USA	CT, ERCP, EUS
8. Jiang H et al. [67]	2018	Canada	US, MRI, CT, ERCP
9. Lin T et al. [41]	2020	Taiwan	CT, MRCP
10. Chu C et al. [34]	2019	China	CT
11. Ohtani M et al. [68]	2021	Japan	18F- FDG PET/CT
12. Matsubayashi H et al. [73]	2021	Japan	CT
13. Cho M et al. [79]	2018	South Korea	CEUS
14. Srisajjakul S et al. [80]	2020	Thailand	CT and MRI
15. Bieliuniene E et al. [71]	2020	Lithuania	CT- and MRI
16. Tacelli M et al. [81]	2022	Italy	EUS
17. Dickerson L et al. [48]	2019	UK	CT
18. Miyoshi H et al. [74]	2019	Japan	CT, MRI, FDG-PET
19. Harmsen F et al. [82]	2018	Germany	MDCT, B-mode EUS, ESE, CELMI-EUS, EUS-FNA
20. Zhu L et al. [83]	2021	China	MR elastography and tomoelastography
21. Wyse J et al. [84]	2018	Canada	EUS
22. Luo B et al. [44]	2019	China	ERCP
23. Zhang H et al. [58]	2018	China	CT
24. Grassia R et al. [61]	2020	Italy	EUS-FNB and EUS-FNA
25. Konings I et al. [85]	2018	Netherlands	EUS
26. Enjuito D et al. [86]	2021	Spain	CT
27. Jeon C et al. [87]	2020	USA	CT
28. Kim H et al. [69]	2022	South Korea	CT and MRI
29. Teske C et al. [88]	2022	Germany	infrared spectroscopy
30. Hsu W et al. [65]	2018	Taiwan	CT, MRI, FDG-PET, EUS
31. Mohamed A et al. [89]	2017	France	CT
32. Rana S et al. [64]	2018	India	CT, MRI, FDG-PET, EUS, CE-EUS
33. Dite P et al. [19]	2019	Czech Republic	US, EUS
34. Walling A et al. [63]	2017	USA	CT, MRCP, EUS, ERCP
35. Rashid S et al. [90]	2018	India	FDG-PET, EUS
36. Tirkes T et al. [91]	2019	USA	CT and MRI, and MRCP
37. Ma X et al. [92]	2022	China	Enhanced CT Radiomics
38. Ishikawa T et al. [37]	2020	Japan	EUS-FNA, CE-EUS, CT
39. Bartell N et al. [93]	2019	USA	EUS-FNA, MRI, CT
40. Macinga P et al. [31]	2017	Czech Republic	CT, EUS, ERCP
41. Liu Y et al. [94]	2020	China	US, CEUS
42. Guo T et al. [95]	2021	China	EUS
43. Konur S et al. [96]	2020	Turkey	CT
44. Marinho R et al. [21]	2019	Portugal	CT, ERCP
45. Li G et al. [16]	2019	China	CT, MRI, FDG-PET
46. Detlefsen S et al. [38]	2018	Denmark	EUS

## Data Availability

This is a review of published research. No new data were generated. The full list of articles reviewed is under References. Alternatively, this information may be requested from the corresponding author in writing.

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
