# Peer review of "Diagnostic Differentiation between Pancreatitis and Pancreatic Cancer: A Scoping Review"

_diagnostics, 2024, doi:10.3390/diagnostics14030290_

Round 1

Reviewer 1 Report

Comments and Suggestions for Authors

In this review, the authors summarized recent publications (2017-2022) on the diagnostic differentiation between pancreatitis and pancreatic cancer while identifying knowledge gaps in the field. However, accurately distinguishing between the two conditions remains a challenge, particularly when a mass is present in the head of the pancreas. Notwithstanding advancements in understanding the association between pancreatitis and pancreatic cancer, including the correlation between histopathological grading systems, non-invasive imaging techniques, and biomarkers in chronic pancreatitis to determine the risk of progression to pancreatic cancer, or differentiating between the two conditions, several knowledge gaps exist. Further research is needed to enhance our understanding of these aspects. Although this review has some limitations, it is helpful in clinical settings because patients may benefit from diagnosis and management of pancreatitis and pancreatic cancer. Overall, there are three minor issues to be addressed before its acceptance.

Minor comments:

1.     The authors should add the appropriate contents in square brackets in Figure 2.

2.     The authors should confirm the Prisma flow chart in Figure 5 is shown correctly and accurately. 

3.     The authors should confirm all figure labels are correctly cited in the right places of the whole manuscript.

Author Response

Responses to Reviewer #1

  1. The appropriate contents for figure 1 are/were in the first and revised version of the manuscripts submitted on the 10th December 2023 in response to the queries of your correspondence of the 7th December 2023. In your correspondence of the 7th December 2023, we were requested to clarify potential copyright infringements for some of the cited figures. In our response on the 10th December 2023, we opted to remove figures that did not have copyright permission and opted to retain the citation of the text illustrated by the diagrams only. We also submitted a revised order of figures and labels. It is possible that due to errors following editing or incompatible software, the content may have been inadvertently removed. I have re-entered the content with this revision of the manuscript.
  2. The Prisma-ScR flow chart in figure 2, contained minor typographical errors which I have corrected and submitted a revised figure 2 diagram in pdf as per requested.
  3. All figures and labels as found in our revised manuscript and revised figures in our correspondence dated 10thDecember 2023 are correctly cited. In your correspondence of the 7th December 2023, we were requested to clarify potential copyright infringements for some of the cited figures. In our response on the 10th December 2023, we opted to remove figures that did not have copyright permission and opted to retain the citation of the text illustrated by the diagrams only. We also submitted a revised order of figures and labels. It appears that there was an unfortunate error resulting in the original manuscript (instead of the revised on of the 10th December 2023) being send to reviewer #1. We have made corrections to the manuscript that will be submitted with this correspondence.

Reviewer 2 Report

Comments and Suggestions for Authors

The authors present a review regarding the diagnostic differentiation between pancreatitis and pancreatic cancer. In my opinion it is a good summary, however, does not add valuable information for the clinicl practice. 

In my opinion the biggest problems in this field are  1/ differentiation  of CP with tumor( is it neoplastic or inflammatory mass). Adding a diagnostic algorithm suggesting the managment of EUS with negative biopsy in such case ( How many  times should we biopsy the tumor with EUS? what else can we do when the biopsy is negative?) would be valuable for the readers. These cases are often also surgical challenge: inflammation, collateral circulation.....

2/ Diagnosis of AIP : in which cases of CP,AP,Pancreatic tumors should we suspect aotimmune pancreatitis  

Author Response

Responses to reviewer #2

  1. We agree that a diagnostic algorithm for chronic pancreatitis with a mass would be useful and would add to the current literature. However, this is not concordant with the aims of the study and would be beyond the scope of a scoping review (Arksey). This review aimed to map the current literature that addresses tumour markers and imaging modalities used to differentiate chronic pancreatitis from pancreatic cancer and to identify research gaps on the topic. It was our belief that proposing a diagnostic algorithm based on results from a scoping review would exceed the requirement of a scoping review. Unlike systematic reviews, scoping reviews map the literature without critically appraising it (Levac, Munn). Existing literature would need to be critically appraised before using it as a guide to a diagnostic algorithm. Notwithstanding this concern, we have added a supplementary figure 5 to the manuscript as our proposed diagnostic algorithm for CP with head of pancreas mass.
  2. The diagnosis of Auto-immune pancreatitis remains challenging. This is due to a lack of randomised control or case control studies addressing the diagnostic accuracies of available modalities. Serum IgG4 is still the marker with the highest accuracy, but it’s utility combined with imaging should be explored. This is a research gap that we have added to the paper.

References

  1. Arksey H, O'Malley L. Scoping studies: towards a methodological framework. International journal of social research methodology. 2005 Feb 1;8(1):19-32.
  2. Levac D, Colquhoun H, O'Brien KK. Scoping studies: advancing the methodology. Implementation science. 2010 Dec;5:1-9.
  3. Munn Z, Peters MD, Stern C, Tufanaru C, McArthur A, Aromataris E. Systematic review or scoping review? Guidance for authors when choosing between a systematic or scoping review approach. BMC medical research methodology. 2018 Dec;18:1-7.

Round 2

Reviewer 2 Report

Comments and Suggestions for Authors

I 'm pleased with the answers to my remarks

Author Response

Please see the pdf attached.
